# Liquid–Liquid Phase Separation Sheds New Light upon Cardiovascular Diseases

**DOI:** 10.3390/ijms242015418

**Published:** 2023-10-21

**Authors:** Ziyang Cai, Shuai Mei, Li Zhou, Xiaozhu Ma, Qidamugai Wuyun, Jiangtao Yan, Hu Ding

**Affiliations:** 1Division of Cardiology, Departments of Internal Medicine, Tongji Hospital, Tongji Medical College, Huazhong University of Science and Technology, Wuhan 430030, China; m202376476@hust.edu.cn (Z.C.); moshine@hust.edu.cn (S.M.); m202276233@hust.edu.cn (L.Z.); xzma2023@hust.edu.cn (X.M.); d202282253@hust.edu.cn (Q.W.); 2Hubei Key Laboratory of Genetics and Molecular Mechanisms of Cardiological Disorders, Wuhan 430030, China; 3Genetic Diagnosis Center, Tongji Hospital, Tongji Medical College, Huazhong University of Science and Technology, Wuhan 430030, China

**Keywords:** liquid–liquid phase separation, biomolecular condensates, cardiovascular diseases

## Abstract

Liquid–liquid phase separation (LLPS) is a biophysical process that mediates the precise and complex spatiotemporal coordination of cellular processes. Proteins and nucleic acids are compartmentalized into micron-scale membrane-less droplets via LLPS. These droplets, termed biomolecular condensates, are highly dynamic, have concentrated components, and perform specific functions. Biomolecular condensates have been observed to organize diverse key biological processes, including gene transcription, signal transduction, DNA damage repair, chromatin organization, and autophagy. The dysregulation of these biological activities owing to aberrant LLPS is important in cardiovascular diseases. This review provides a detailed overview of the regulation and functions of biomolecular condensates, provides a comprehensive depiction of LLPS in several common cardiovascular diseases, and discusses the revolutionary therapeutic perspective of modulating LLPS in cardiovascular diseases and new treatment strategies relevant to LLPS.

## 1. Introduction

Intracellular biochemical reactions require complex spatiotemporal regulations. Diseases may develop and progress when various important biological activities in cells are disrupted, owing to disordered spatiotemporal regulation. Cells possess intracellular compartments that regulate biological processes with high precision and extreme dynamics [1]. Some of these compartments are familiar organelles, including the Golgi apparatus and endoplasmic reticulum, which have a typical lipid bilayer as a barrier and create a distinctive biochemical environment. However, some compartments do not possess lipid bilayers. These unique compartments, lacking a physical barrier but separated from the surrounding environment via liquid–liquid phase separation (LLPS), are termed membraneless organelles or biomolecular condensates. Biomolecular condensates intervene in diverse key cellular activities, including transcription [2], intracellular signaling [3], chromatin organization [4], mitosis [5], and ferroptosis [6].

Cardiovascular diseases are the leading cause of death worldwide [7,8,9]. However, understanding of the pathogenesis of cardiovascular diseases remains limited [10,11,12,13,14,15]. Therefore, there is an urgent need to develop novel concepts to comprehensively understand the course of cardiovascular diseases and identify effective therapeutic targets. Although a few studies have demonstrated an association between LLPS and cardiovascular diseases [16,17,18], many cardiologists remain unaware of the broad potential of LLPS in cardiovascular diseases. In this review, we aimed to elucidate the mechanisms and functions of LLPS comprehensively, discuss its potential roles in the pathological process and treatment of cardiovascular diseases, highlight the significance of LLPS for cardiologists, and provide them with insights and advancements that can be achieved.

## 2. LLPS

LLPS is a biophysical process in which macromolecules assemble into a condensed phase and separate from the surrounding solution [19] (Figure 1). The condensed phase is a circular droplet inside the dilute phase because of surface tension [20]. To help understand the concept of LLPS, an analogy of a mixture of oil and vinegar can be considered. When oil and vinegar are mixed, separation between the oil and vinegar phases is observed [21].

### 2.1. Multivalency Drivers of LLPS

Multivalency is the ability to establish multiple interactions with other molecules [22] and is the primary characteristic of both multiple folded domains and intrinsically disordered regions of the protein [23,24] (Figure 2). LLPS is driven by weak multivalent interactions among biomolecular condensate components [19,25]. When the nonspecific multivalent weak interaction between the macromolecules surpasses that of the macromolecule–solvent, the solution separates into two distinct phases [26]. Consequently, macromolecules become concentrated in the condensed phase, which may accelerate the chemical reactions between them [27,28].

#### 2.1.1. Multiple Folded Domains

Proteins with multiple folded domains have several binding sites, each capable of binding to multiple binding partners [29]. The multivalency provided by multiple folded domains enables proteins to undergo phase separation. The first multivalent system discovered was nephrin-NCK-N-WASP, in which every component has multiple identical binding domains. These domains are cross-linked, and their interactions result in LLPS [30] (Figure 2A). Additionally, the oligomerization of NPM1 endows it with multiple identical domains that bind to proteins with Arg-rich linear motifs, promoting the LLPS of NPM1 [31] (Figure 2B). Summarily, both examples illustrate multimodal interactions as potential drivers of LLPS.

#### 2.1.2. Intrinsically Disordered Regions (IDRs)

The multivalency of IDRs is represented by π–π, charge–charge, and cation–π interactions [29] (Figure 2C). These interactions among IDRs promote the association of proteins containing IDR with each other while excluding other biomolecules, consequently forming a liquid-like compartment fused with specific proteins. IDRs are protein segments that lack a defined tertiary structure and generally contain biased amino acid distributions [32,33]. Similarly, IDRs are characterized by a smaller proportion of hydrophobic residues and a larger proportion of hydrophilic and charged residues [34]. These distinct characteristics are vital for IDR-mediated LLPS [32]. Hence, they are widely used to identify IDRs in proteins and predict the propensity for LLPS. Considering the close relationship between IDRs and LLPS, predictors and databases focusing on IDRs and intrinsically disordered proteins (IDPs) have been set up, which benefit researchers in evaluating the possibility of their target protein for LLPS. Table 1 shows a list of these tools for further reference.

The IDR is a key driver of phase separation. For instance, the removal of the N-terminal IDR of Nur77 significantly impaired its ability to form droplets [44]. Moreover, the IDR of some proteins can spontaneously assemble into condensates. For example, the N-terminal IDR of Ddx4 can self-associate into biomolecular condensates both in vitro and in vivo [45].

Notably, alterations in the amino acid sequence of IDR affect LLPS. The phase separation of HMGB1 is altered when the intrinsically disordered region of HMGB1 is replaced by an arginine-rich region [46]. Furthermore, this replacement causes nucleolar HMGB1 mispartitioning and nucleolar dysfunction [46]. Functional assessments of nine other transcription factors revealed a strong correlation between the degree of protein mispartitioning into the nucleolus and the proportion of arginine residues within the IDR created by frameshift mutation [46].

### 2.2. Physiological Regulation of LLPS

Intracellular LLPS is subject to multilevel modification [47]. Any subtle alterations that affect the composition of condensates and multivalent interactions may influence LLPS, including changes in environmental factors and post-translational modifications (PTMs) [32].

#### 2.2.1. Physical Parameters

Changes in physical parameters affect the thermodynamics of the condensate and the multivalent interactions among biological macromolecules. For LLPS, it is crucial to consider physical factors, including salt concentration, temperature, pH, and pressure. For example, lowering salt concentration and temperature induces LLPS in hnRNPA1 [48,49]. Additionally, stress-triggered LLPS of poly(A)-binding protein is pH-dependent [50]. Besides, Cina et al. demonstrated that excessively high pressure causes the concentrated and dilute phases of the γ-crystallin system to mix into a homogeneous phase [51]. However, phase separation recurs after decreasing the pressure [51]. Moreover, pressures of approximately 3 kbar do not affect the secondary structure of γ-crystallin [51], suggesting that high pressure regulates multivalent interactions.

Apart from environmental factors, the concentration of biological macromolecules plays a critical role in LLPS [52]. Weber et al. found that nucleoli failed to assemble at concentrations below the threshold concentration of fibrillarin-1 (FIB-1) [53]. Nevertheless, at concentrations above the threshold, a higher concentration of FIB-1 resulted in the formation of larger nucleoli via phase separation [53].

In addition to the conventional physical parameters, recent findings have established a relationship between LLPS and unexpected factors. Surface tension is a physical property associated with biomolecular condensates [20]. Wang et al. revealed that MLX, MYC, and inositol polyphosphate multikinase act as surfactants in a synergistic manner to reduce the surface tension of transcription factor EB (TFEB) condensates, which in turn affects the affinity of the condensates to DNA, leading to the negative regulation of the autophagy lysosome pathway (ALP) [54]. However, this is no exception. The same researchers also discovered that RUNX3 and HOXA4 play similar roles in TAZ-TEAD4 condensates [54]. These remarkable results indicate the possibility of targeting LLPS by designing surfactants that interfere with the condensate function. Recently, the surprising effect of magnetic fields on LLPS has been reported. Lin et al. observed that magnetic fields disrupt the LLPS of Tau-441 and reduce apoptosis in vitro [55]. Therefore, we can envision a novel strategy to modify the function of LLPS via physical strategies.

#### 2.2.2. PTMs

PTMs, including phosphorylation, ubiquitination, methylation, and acetylation, have been discovered to participate in the regulation of biomolecular condensates [56,57,58,59,60]. First, PTMs in the IDRs of the condensate components can alter phase separation. For instance, droplet formation is impaired in both Ddx4 and hnRNPA2 due to arginine methylation of their IDRs [45,60]. Similarly, Saito et al. identified a negative impact of lysine acetylation in the IDR of LLPS [59]. They demonstrated that the acetylation of a single lysine residue on DDX3X-IDR is sufficient to impair LLPS, and that the acetylation of poly-lysine residues has a more severe impact [59]. Conversely, PTMs may also contribute to LLPS. Zhang et al. showed that heat-shock transcription factor 1 (HSF1) is phosphorylated during heat shock and accumulates at heat shock protein (HSP) gene loci to form condensates that segregate the transcription machinery and promote HSP gene expression [61]. Briefly, the phosphorylation of HSF1 is necessary for LLPS.

Second, PTMs can modify the composition and function of biomolecular condensates. Phosphorylation of Pol II is regarded as a perfect illustration. The low-phosphorylated Pol II C-terminal domain (CTD) is incorporated into the mediator condensate, facilitating translation initiation [62]. However, when CDK7 or CDK9 phosphorylates the CTD of Pol II, the Pol II-containing condensate is converted into a splicing factor condensate that participates in RNA splicing [62].

Finally, PTMs may create the multivalent interactions necessary for LLPS. For example, Du et al. demonstrated that the binding of polyubiquitin chains to nuclear factor kappa B (NF-kB) essential modulator (NEMO) promotes its assembly into condensates [63]. Moreover, they found that disease-associated mutations in the ubiquitin-binding domains of NEMO affected the formation of condensates [63], proving that the multivalent interaction of polyubiquitin chains with NEMO is essential for LLPS.

Intriguingly, these findings reveal the diverse roles of PTMS in LLPS, and the modulation of PTMs appears to be a highly feasible strategy for inhibiting LLPS and its pathological consequences.

#### 2.2.3. RNA

RNA is a constituent of specific biomolecular condensates [64]. Nevertheless, RNA can also modulate the properties and functions of biomolecular condensates rather than just being a component [65]. For example, RNAs enhance the liquidity of PGL granules and inhibit EPG-2 recruitment to prevent their degradation [65].

Long non-coding RNA (lncRNAs) are important regulators of cellular signal transduction [66] and are mediators of LLPS [67,68,69]. Daneshvar et al. demonstrated that multivalent interactions between the lncRNA DIGIT and BRD3 promote the formation of liquid-like RNA-BRD3 condensates [67]. Likewise, lncRNA NEAT1 possesses multiple binding sites for TDP43 and serves as a scaffold for TDP43 condensates [68]. Indeed, lncRNA NEAT1 co-phases with TDP43 to form condensates, and the stress-induced increase in lncRNA NEAT1 levels promotes the LLPS of TDP43 [68]. Although lncRNAs have not been reported to modulate phase separation in cardiovascular diseases, the therapeutic potential of lncRNAs that interfere with LLPS is still worth exploring. Further studies are required to identify cardiovascular disease-related lncRNAs that participate in LLPS. Furthermore, a circular RNA (circRNA), circASH2, has been reported to enhance the LLPS of YBX1 [70]. Therefore, the possibility of uncovering various RNAs involved in cardiovascular disease-related LLPS should not be neglected.

#### 2.2.4. Adenosine Triphosphate (ATP)

ATP is a cellular regulator of LLPS under physiological conditions [71,72]. On the one hand, Patel et al. revealed that relatively high concentrations of ATP in cells inhibit the LLPS of IDPs [73]. The charged and hydrophobic moieties of ATP interact with the amino acids in IDP, enhancing protein solubility and preventing the formation of condensates [73]. On the other hand, ATP hydrolases facilitate condensate assembly via ATP hydrolysis [74]. Hondele et al. demonstrated in vitro and in vivo that DEAD-box ATPases (DDXs) promote phase separation by binding to ATP [75]. However, ATP hydrolysis leads to the release of RNA from condensates, affecting multivalent interactions between DDXs and RNA and resulting in the disassembly of membraneless organelles [75]. Although there is evidence for the role of ATP in LLPS, further investigation is still needed.

#### 2.2.5. Chaperones

Molecular chaperones are proteins that assist other proteins in attaining a functional conformation [76]. Several molecular chaperones, including Class I and II Hsp40, HSP27, and HSP70 proteins, have a propensity for LLPS [61,77,78]. In other words, these chaperones undergo LLPS themselves. Chaperone proteins also play a role in the regulation of LLPS [79]. For instance, Liu et al. reported that HSP27 interacts with the IDR of fused-in sarcoma (FUS), subsequently interfering with the intramolecular and intermolecular interactions of FUS proteins [78]. As a consequence, HSP27 inhibits the LLPS of FUS. Similarly, when a heat shock event persists for an extended duration, an abundance of HSP70 disrupts the LLPS of HSF1 and prevents the liquid–gas phase transition of HSF1 by binding to HSF1 [61].

### 2.3. Functions of Biomolecular Condensates

#### 2.3.1. LLPS in Transcriptional Regulation

The phase-separated condensate is involved in the entire process of transcription; therefore, LLPS is of great significance for transcriptional regulation. Boija et al. proposed a model of the transcriptional condensate involved in gene activation [2]. They found that the transcription factor OCT4 activates gene transcription through the condensate formed by interactions between the activation domain of OCT4 and coactivators in the mediator complex [2]. Furthermore, transcription condensates are mediated by strong specific transcription factor-enhancer element interactions and weak multivalent transcription factor-coactivator IDR interactions [80]. Super-enhancers (SEs) are clusters of highly active and sensitive enhancers distributed at high concentrations of transcription factors, coactivators, and RNA Polymerase II [81,82,83]. The LLPS of coactivators concentrates the transcription apparatus on the target genes of super-enhancers, promoting the initiation of gene transcription [84,85]. Therefore, LLPS is a potential mechanism by which super-enhancers promote transcription.

Interestingly, different stages of transcription form distinct transcriptional condensates that differ in composition and function [86]. Changes in the transcriptional condensates are associated with RNA expression [87]. In the early stages of transcription initiation, a low concentration of short RNA transcribed from enhancers and promoters favors the formation of transcriptional condensates [87]. Meanwhile, a high concentration of long RNA during transcriptional bursts causes transcription condensates to disintegrate [87].

In addition to the role of LLPS in transcriptional condensates, it may also inhibit transcription. Chong et al. discovered that a stronger interaction between TAF15 IDR and EWS-IDR contributes to more apparent LLPS, yet reduces the expression of target genes of the EWS::FLI1 fusion protein [88]. This is probably because TAF15 puncta captures EWS::FLI1 and reduces the number of EWS::FLI1 molecules that freely access the target genes [88]. Pathogenic mutations in eleven-nineteen leukemia (ENL) illustrate the dual role of LLPS in transcriptional regulation. Disease-related mutations in ENL have a stronger propensity for LLPS than wild-type mutations [89]. In cellular experiments, ENL mutants at near-endogenous concentrations form condensate at specific gene loci and promote the expression of their target gene, HOXA11 [89]. Conversely, overexpression of ENL mutants results in the formation of larger condensates that do not bind to chromosomes and fail to activate target genes [89].

#### 2.3.2. LLPS in Intracellular Signaling

LLPS may be a key step in intracellular signal transduction. For example, NEMO activates downstream IkB kinase (IKK) and NF-kB signaling via LLPS, whereas the NEMO mutant cannot form condensates and has a diminished ability to activate IKK correspondingly [63]. Besides, in response to TLR4 receptor signaling, TRAF6 undergoes oligomerization to form condensates that promote TRAF6 self-ubiquitination and subsequent NF-kB activation [90].

Biomolecular condensates can function as signaling hubs. As an activation signal for the Hippo signaling pathway (a central pathway that controls tissue growth, homeostasis, and regeneration), osmotic stress induces the formation of angiomotin (AMOT)/kidney and brain protein (KIBRA)/sarcolemma-associated protein (SLMAP) condensates, which then recruit cascade kinases to activate the signaling pathway [3]. Unexpectedly, SLMAP condensates that inhibit the Hippo pathway lose their inhibitory effect after co-phasing with AMOT/KIBRA condensates [3]. Significantly, this discovery provides a novel paradigm for disrupting the function of biomolecular condensates through multiphase coalescence rather than through dissolving condensates [3].

#### 2.3.3. LLPS in Chromatin Organization

LLPS, an intrinsic property of chromatin [91], plays a pivotal role in regulating chromatin’s structure. For example, heterochromatin is an inactive region of the eukaryotic genome [92], and its formation is driven by LLPS [93]. Moreover, transcription factors regulate the 3D structure of chromatin through LLPS. Wang et al. observed that the transcription factor OCT4 forms chromosomal loops through LLPS, and regulates TAD remodeling [94]. A recent study revealed that long-distance chromatin interactions are mediated by LLPS [95]. The RING1 and YY1 binding protein (RYBP) facilitates the LLPS of CCCTC-binding factor (CTCF), which regulates the interactions between the A compartments of the chromosome [95].

#### 2.3.4. LLPS in DNA Damage Repair

DNA damage repair (DDR) is a cascade of signals that mediate DNA damage detection and repair [96]. Indeed, LLPS is one of the molecular mechanisms of DDR [97]. Crosstalk between DDR and LLPS maintains genomic stability [96].

A variety of dynamic DNA repair compartments generated by DDR exist in cells and can integrate the aggregation of repair factors with the activation of signaling factors [98]. Kilic et al. demonstrated that the key repair protein, 53BP1, condenses at DNA damage regions and activates p53 to protect the break site [98]. Moreover, phase separation of SUMOylated ring finger protein 168 (RNF168) inhibits DNA damage repair [99]. RNF168 condensates impede RNF168 recruitment at DNA damage sites and segregate the downstream protein, 53BP1 [99]. Additionally, Frattini et al. found that TopBP1 condensates could amplify the effect of ataxia-telangiectasia-mutated and Rad3-related (ATR) on phosphorylating checkpoint kinase 1 (Chk1), leading to the blockade of DNA replication strands [100].

#### 2.3.5. LLPS in Autophagy

Autophagy is an autophagosome-mediated process in eukaryotic cells that transports cellular components to lysosomes for degradation [101]. LLPS plays a critical part in autophagosome biosynthesis [102]. Fujioka et al. found that the pre-autophagosomal structure that drives autophagy is a condensate formed by LLPS [103]. Moreover, biomolecular condensates are involved in delivering misfolded proteins to autophagosomes. For example, during selective autophagy, misfolded proteins tagged with ubiquitin assemble into p62 condensates, resulting in the concentration and sequestration of misfolded proteins [104]. Subsequently, these concentrated misfolded proteins are engulfed and degraded by autophagosomes [104]. Furthermore, biomolecular condensates are specifically recognized by autophagosome receptors and cleared by autophagosomes [101].

## 3. Potential Roles of LLPS in Cardiovascular Diseases

LLPS is associated with various biological activities occurring in all human body cells. Therefore, it is reasonable to speculate that aberrant biological activity caused by LLPS may contribute to the pathogenesis of cardiovascular diseases. Of note, existing literature has demonstrated that LLPS plays a significant role in several cardiovascular diseases (Figure 3).

### 3.1. LLPS in Cardiac Fibrosis

Cardiac fibrosis is a pathological process involving cardiac remodeling characterized by a distorted cardiac structure and impaired cardiac function due to the excessive deposition of collagen and extracellular matrix (ECM) proteins [105,106]. Under pathological conditions, cardiac fibroblasts are activated and differentiate into myofibroblasts to mediate cardiac fibrosis [107]. Horii et al. discovered that the mechanosensitive protein VGLL3 is specifically expressed in cardiac myofibroblasts, and confirmed that VGLL3 undergoes LLPS via its IDR to promote cardiac fibrosis [17]. Furthermore, VGLL3 colocalizes with non-paraspeckle NONO condensates and suppresses the production of miR-29B, which disturbs the expression of collagen and several fibrotic molecules [17]. Thus, it is probable that the LLPS of VGLL3 is vital for the biogenesis of miRNAs, which further regulate cardiac fibrosis. Additionally, BRD4, a protein with LLPS propensity, has been identified as an efficient target for attenuating cardiac fibrosis [108,109]. BRD4 is a transcription coactivator enriched in super-enhancers (SEs), and has the capacity to undergo phase separation through its IDR [84]. Stratton et al. discovered a substantial increase in the concentration of BRD4 at the promoter region and several distinct proximal enhancers of Sertad4 in cardiac fibroblast after transforming growth factor β (TGF-β) stimulation [110]. Subsequently, knockdown of Sertad4 markedly impaired cardiac fibroblast activation induced by TGF-β [110]. Therefore, it is promising that the LLPS of BRD4 mediates the activation of cardiac fibrosis-related genes. Undoubtedly, these findings offer a new perspective on therapeutic targets for cardiac fibrosis other than the fibrogenic signaling pathways [111].

### 3.2. LLPS and Heart Failure

Heart failure is a clinical syndrome characterized by cardiac systolic and/or diastolic dysfunction rather than a specific disease [112]. With remarkable advances in genetics, the integration of Mendelian and complex genetics has unveiled the prospects of precision medicine for heart failure [113]. In this case, LLPS may be a revolutionary target of precision medicine for heart failure, as Xie et al. found that the inhibition of the LLPS of RUNX family transcription factor 2 (Runx2) can reverse pathological cardiac remodeling due to heart failure [18]. In the nucleus of cardiomyocytes, the Runx2 condensate colocalizes with MED-1 (a subunit of the transcriptional mediator complex) on the epidermal growth factor receptor (EGFR) promoter and promotes EGFR expression [18], suggesting that Runx2 is incorporated into the transcriptional condensate via LLPS. Previously, pharmacological intervention with LLPS in cancer has been reported to inhibit tumor growth, metastasis, and chemotherapy resistance [114,115]. However, since the role of LLPS in the failing heart has been discovered, it is imperative to focus on other transcription factors capable of LLPS in cardiovascular diseases, and to explore their underlying regulatory mechanisms.

### 3.3. LLPS in Dilated Cardiomyopathy (DCM)

DCM is a heterogeneous nonischemic cardiac muscle disease characterized by abnormal left ventricular or biventricular dilation and continuous contractile dysfunction [116,117]. Moreover, the heterogeneous etiology of DCM comprises genetic and nongenetic factors [118,119]. RNA-binding Motif-20 (RBM20) is an RNA-binding protein mainly distributed in the nucleus, and some of its mutations cause DCM [117]. Schneider et al. demonstrated that the misguided phase separation of RBM20 mutants is associated with the abnormal accumulation of ribonucleoprotein (RNP) granules in the sarcoplasm, resulting in cardiomyopathy [16]. Under physiological conditions, the tubulin network integrates the transportation of mRNA from the nucleus to the Z-disc [120]. However, RNP granules incorporating pathological RBM20 mutants assemble abnormally at the Z-disc of the cytoskeleton and throttles the transportation of genetic information within the tubulin network [16]. Moreover, dysregulated RNP granules capture Actin alpha cardiac muscle 1 (a critical component of the cardiomyocyte cytoskeleton) at their phase boundary, thus disrupting normal actin polymerization (a key cellular process for force production in cardiomyocytes) [16,121]. Similarly, many RNA-binding proteins (RBPs) have multivalent modular structures that enable them to undergo LLPS [122,123,124]. Importantly, several hundred RBPs have been found to be relevant to cardiac function and cardiovascular diseases [125,126,127]. Therefore, we believe that additional RBPs that cause cardiovascular diseases via LLPS can be identified as effective therapeutic targets for modulating LLPS and disrupting LLPS-mediated pathological processes.

### 3.4. LLPS in Atherosclerosis

Atherosclerosis is characterized by the accumulation of lipids, ECM, inflammatory cells, smooth muscle cells, and necrotic cell debris in the innermost layer of the arteries [128]. It is associated with dysregulated autophagy, inflammation, and abnormal lipid metabolism [129,130]. Li et al. showed that LLPS in macrophages participates in atherosclerotic pathogenesis [131]. Oxidized low-density lipoprotein (Ox-LDL) inhibits the cytoplasmic nuclear transport of TFEB, which in turn inhibits the activation of lysosomal genes via TFEB condensates in the nucleus, leading to autophagy deficiency [131,132,133]. Subsequently, autophagy defects result in elevated reactive oxygen species (ROS) levels and P300 activity [131]. Consequently, the LLPS of P300 recruits BRD4 to the promoter region of inflammatory genes, facilitating the expression of inflammatory factors [131,134]. Clearly, LLPS plays a crucial role in ox-LDL-induced autophagy defects and inflammation, thereby contributing to atherogenesis. We expect that the other pathological mechanisms of atherosclerosis associated with LLPS will be revealed in the near future, and will aid in novel drug design.

## 4. Discussion and Conclusions

Recent studies have demonstrated that biomolecular condensates cooperate to regulate various biological processes. The dysregulation of key biological processes is likely to induce cardiovascular diseases; therefore, LLPS may be an underlying pathological mechanism in cardiovascular diseases. Although not many studies have demonstrated a clear relationship between phase separation and cardiovascular diseases, further investigation of phase separation promises to bring revolutionary advancements in understanding the mechanism and treatment of cardiovascular diseases. Thus, it is crucial to spare no effort in identifying dynamic condensates in cells that correlate with cardiovascular diseases, and to further explore the mechanisms and functions of these condensates.

Additionally, the potential of condensate-modifying therapeutics in cardiology is enormous [135]. For instance, PTMs and RNAs may disrupt the LLPS of some pathological condensates in cardiovascular diseases [135]. Furthermore, small molecules that modify the material properties of condensates have been developed and proven to compromise the function of LLPS [136]. Moreover, integrating LLPS with other novel technologies provides a more effective tool for exploring biological condensates and applying gene therapies in cardiovascular diseases. For example, proteolysis-targeting chimeras (PROTAC) can rapidly and reversibly regulate target proteins in biomolecular condensates. Therefore, Shi et al. devised the BRD4 PROTAC, which makes it possible to observe the dynamics of other condensate components under the continued disruption of BRD4 condensates [137]. This technique can potentially help researchers to study the functions of transcription factors with LLPS properties in cardiovascular diseases. Furthermore, the feasibility of using phase-separating proteins to enhance dCas9-VPR activity has been demonstrated. Liu et al. fused the human NUP98 and FUS IDR domains to dCas9-VPR, and observed a substantial increase in the efficiency of transcriptional activation [138].

Generally, the cutting-edge findings of LLPS have primarily focused on physiological and pathological processes, whereas the clinical application of LLPS in cardiovascular diseases remains to be explored by the efforts of cardiologists. With the collaboration of biologists and cardiologists, we believe the mystery of phase separation in cardiovascular diseases can be unraveled, and additional cardiovascular disease-related condensates identified.

## Figures and Tables

**Figure 1 ijms-24-15418-f001:**
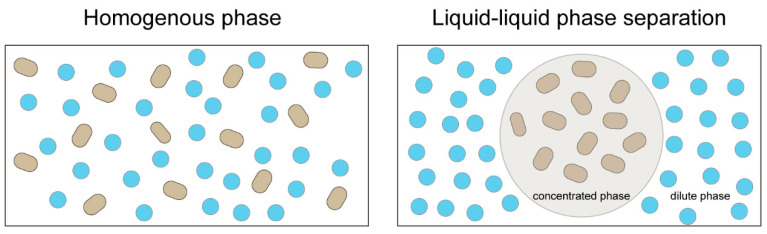
A schematic diagram of liquid–liquid phase separation. (**Left**) In a homogenous phase, molecules are distributed uniformly. (**Right**) When specific macromolecules undergo LLPS, they assemble into a concentrated phase, while separating themselves from the surrounding dilute phase.

**Figure 2 ijms-24-15418-f002:**
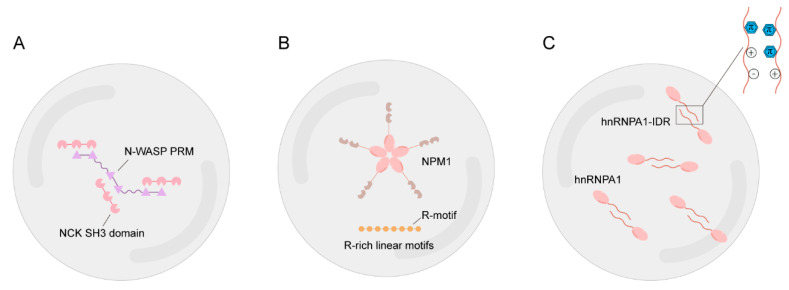
Molecular mechanisms of multivalency that drive LLPS. (**A**). Proteins with multiple folded domains interact multivalently with each other. N-WASP has six binding sites for NCK, and the NCK SH3 domain has three binding sites for N-WASP PRM. Therefore, N-WASP interacts with NCK and other proteins multivalently to undergo LLPS. (**B**). NPM1 oligomerization enables it to have ten binding sites for R motif; thus, it interacts multivalently with R-rich linear motifs to form a biomolecular condensate. (**C**). Multivalent interactions between IDRs include π–π, charge–charge, cation–π interactions. These interactions between hnRNPA1-IDRs mediate hnRNPA1 LLPS.

**Figure 3 ijms-24-15418-f003:**
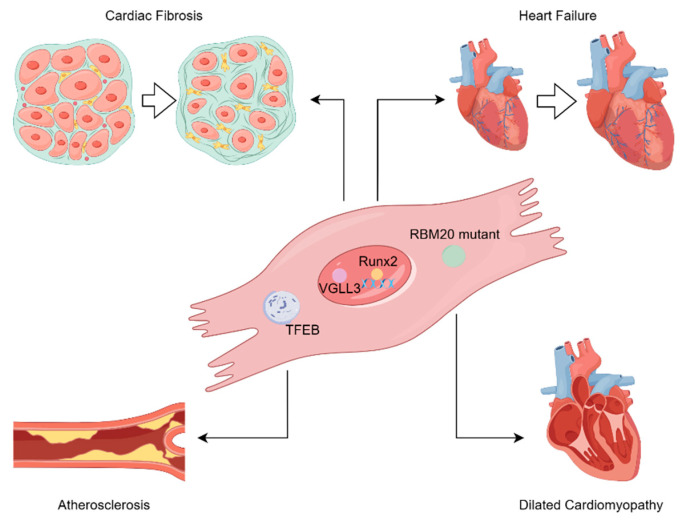
Schematic representation of LLPS in various cardiovascular diseases. (By Figdraw.). (**Top left**) VGLL3 in non-paraspeckle NONO condensates undergoes LLPS and promotes collagen expression, leading to cardiac fibrosis. (**Top right**) LLPS of Runx2 on the promoter of EGFR gene promotes EGFR transcription, thus contributing to heart failure. (**Bottom left**) Disruption of TFEB impairs autophagy and promotes atherosclerosis. (**Bottom right**) The RBM20 mutant assembles into dysregulated biomolecular condensates along cytoskeletal elements, which promotes dilated cardiomyopathy.

**Table 1 ijms-24-15418-t001:** Tools that we use to investigate LLPS and membraneless organelles.

Tool	Type	Characteristic	Availability	Ref.
IUPred3	Predictor	Prediction of IDRs within proteins	https://iupred.elte.hu (accessed on 20 August 2023)	[35]
D2P2	Predictor	Comparisons of IDPs prediction methods	https://d2p2.pro (accessed on 20 August 2023)	[36]
MobiDB	Predictor	Annotation and prediction of intrinsically disordered proteins	https://mobidb.org (accessed on 20 August 2023)	[37]
PhaSePro	Database	Detailed information of LLPS driver proteins, including their post-translational modifications and alternative splicing events known to influence LLPS	https://phasepro.elte.hu (accessed on 20 August 2023)	[38]
PhaSepDB	Database	Comprehensive collection of phase-separation related proteins and detailed annotation for all LLPS entries	http://db.phasep.pro (accessed on 20 August 2023)	[39]
LLPSDB	Database	Collection of LLPS related proteins and their corresponding phase separation conditions	http://bio-comp.org.cn/llpsdb (accessed on 20 August 2023)	[40]
DrLLPS	Database	Collection of 987 regulators of LLPS and 8148 potential client proteins	http://llps.biocuckoo.cn/ (accessed on 20 August 2023)	[41]
DisPhaseDB	Database	Collection of disease related variations in LLPS proteins	http://disphasedb.leloir.org.ar (accessed on 20 August 2023)	[42]
MloDisDB	Database	Link between membraneless organelles and diseases	http://mlodis.phasep.pro (accessed on 20 August 2023)	[43]

## Data Availability

Not applicable.

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
