# Peer review of "Liquid–Liquid Phase Separation Sheds New Light upon Cardiovascular Diseases"

_ijms, 2023, doi:10.3390/ijms242015418_

Round 1
Reviewer 1 Report
The review by Ziyang Cai et al. focuses on liquid-liquid phase separation (LLPS) and its implications in cardiovascular diseases. The manuscript is clearly written and LLPS is well explained at the beginning of the paper. I have only a few minor comments.
1) On line 26, the authors state that the review “elucidates” the underlying pathological mechanism. I think that “elucidates” is an overstatement. This review article does not provide any new experimental data. Please tone down and find a more appropriate word.
2) On line 100, the authors state that IDRs are unstable three-dimensional structural domains containing repetitive sequences. This is not a correct definition of IDRs. Not all IDRs contain repetitive sequences and a conformational ensemble of an IDR cannot be called a 3D structure. Many reviews were written about IDRs. Please find more accurate description of IDRs.
3) Some terms that appear in the text need a brief explanation. Specifically the following:
Hippo pathway (line 265)
RYBP and CTCF (line 279)
RNF168 (line 289)
SE (line 332)
Runx2 (line 345)
4) On line 291 reference to Frattini: Frattini is not a sole author. Please, correct to Frattini et al.
5) On line 401, the authors state “we identified”. This is again an overstatement because the research in ref. 126-129 has been done by research groups other than the authors of this review. Please, find a more appropriate neutral word.
Minor changes required
Author Response
Thank you for your detailed review. Our point-by-point response to your comments is uploaded as a Word file. Please see the attachment.

Reviewer 2 Report
In the current review, the authors have summarized in detail, the mechanisms involved in the Liquid-Liquid Phase Separation of proteins and protein nucleic acid complexes, and how the process can be targeted for the treatment of cardiovascular diseases. The authors should address the following concerns for making the manuscript suitable for publication.
1. The role of LLPS in cardiovascular diseases has been elaborated in a few recent reviews and these are quite similar to the current manuscript.
a) Uversky, V. N. (2023). Liquid–liquid phase separation, membrane-less organelles, and biomolecular condensates in cardiovascular disease. In Droplets of Life, pp. 663-679.
b) Mo Y, Feng Y, Huang W, Tan N, Li X, Jie M, Feng T, Jiang H, Jiang L. Liquid-Liquid Phase Separation in Cardiovascular Diseases. Cells. 2022
c) Vendruscolo, M., Fuxreiter, M. (2022) Protein condensation diseases: therapeutic opportunities. Nat Commun. 13, 5550.
d) Mitrea, D.M., Mittasch, M., Gomes, B.F., Klein, I.A., Murcko, M.A. Modulating biomolecular condensates: a novel approach to drug discovery. Nat Rev Drug Discov. 2022, 21, 841-862.
The authors should highlight the new advances that have been elaborated upon in the current manuscript and are not included in the above reviews.
2. Lines 401-403: The authors have written, ‘Additionally, we identified the potential of condensate-modifying therapeutics in cardiology. For instance, PTMs and RNAs may disrupt the LLPS of some pathological condensates in cardiovascular disease. In the current form, the sentence implies that the authors have found the condensate-modifying therapeutics. The sentence should be reworded with the appropriate citations.
Lines 268: Replace, …..’which previously inhibited’ with ‘that inhibits’
Author Response
Thank you for your constructive review. Our point-by-point response to your comments is uploaded as a Word file. Please see the attachment.
